# TiO_2_ Nanorod-Coated Polyethylene Separator with Well-Balanced Performance for Lithium-Ion Batteries

**DOI:** 10.3390/ma16052049

**Published:** 2023-03-01

**Authors:** Zhanjun Chen, Tao Wang, Xianglin Yang, Yangxi Peng, Hongbin Zhong, Chuanyue Hu

**Affiliations:** 1Modern Industry School of Advanced Ceramics, Hunan Provincial Key Laboratory of Fine Ceramics and Powder Materials, School of Materials and Environmental Engineering, Hunan University of Humanities, Science and Technology, Loudi 417000, China; 2School of Materials Science and Engineering, Dongguan University of Technology, Dongguan 523808, China; 3Western Australia School of Mines, Curtin University, Kalgoorlie, WA 6430, Australia

**Keywords:** TiO_2_ nanorods, polyethylene, ceramic separator, lithium ion batteries, thermal stability

## Abstract

The thermal stability of the polyethylene (PE) separator is of utmost importance for the safety of lithium-ion batteries. Although the surface coating of PE separator with oxide nanoparticles can improve thermal stability, some serious problems still exist, such as micropore blockage, easy detaching, and introduction of excessive inert substances, which negatively affects the power density, energy density, and safety performance of the battery. In this paper, TiO_2_ nanorods are used to modify the surface of the PE separator, and multiple analytical techniques (e.g., SEM, DSC, EIS, and LSV) are utilized to investigate the effect of coating amount on the physicochemical properties of the PE separator. The results show that the thermal stability, mechanical properties, and electrochemical properties of the PE separator can be effectively improved via surface coating with TiO_2_ nanorods, but the degree of improvement is not directly proportional to the coating amount due to the fact that the forces inhibiting micropore deformation (mechanical stretching or thermal contraction) are derived from the interaction of TiO_2_ nanorods directly “bridging” with the microporous skeleton rather than those indirectly “glued” with the microporous skeleton. Conversely, the introduction of excessive inert coating material could reduce the ionic conductivity, increase the interfacial impedance, and lower the energy density of the battery. The experimental results show that the ceramic separator with a coating amount of ~0.6 mg/cm^2^ TiO_2_ nanorods has well-balanced performances: its thermal shrinkage rate is 4.5%, the capacity retention assembled with this separator was 57.1% under 7 C/0.2 C and 82.6% after 100 cycles, respectively. This research may provide a novel approach to overcoming the common disadvantages of current surface-coated separators.

## 1. Introduction

Lithium-ion batteries (LIBs) have attracted extensive attention in recent years due to their balanced electrochemical performance and high energy density. However, with the ever-lasting demand for high-power applications, the safety and reliability of LIBs have become critical. In a lithium-ion battery system, the separator, which functions as the ion conductor and electronic insulation between the anode and the cathode, is of paramount importance for the safety of LIBs [1]. Generally, an ideal separator should possess high porosity and excellent electrolyte wettability for rapid lithium-ion migration as well as desired mechanical strength and toughness for facile manufacturing [2,3]. Currently, the conventional separators for LIBs consist mainly of polyethylene (PE), polypropylene (PP), and their blends, which have suitable mechanical strength, chemical stability, and membrane thickness. However, these separators would suffer from severe thermal shrinkage under abnormal conditions such as overheating or overcharging, resulting in catastrophic thermal runaway, which may cause gas emission, rupture, fire, or explosion [4,5].

In order to enhance the thermal stability of commercial separators, intensive efforts have been made in recent years. On the one hand, alternatives to polyolefin separators (e.g., non-woven separators [6] and solid electrolytes) have been developed. However, the large pore size and poor mechanical properties of non-woven separators severely restrict their further application, and there remain many technical challenges associated with the solid electrolytes, such as interface impedance, processability, and electrode/electrolyte interface stability [7,8,9]. On the other hand, researchers try to improve the thermal stability of polyolefin-based separators by various methods, such as surface grafting, surface coating, blending, and so on. Among them, the surface coating of PE separators with inorganic nanoparticles (e.g., SiO_2_ [10,11,12], Al_2_O_3_ [13,14], metal hydroxides [15,16], zeolite [17,18], ZrO_2_ [19,20] and TiO_2_ [21]) have attracted considerable attention, because it’s an industrially more competitive method to improve the thermal stability of conventional PE separators. However, surface coating of separators with nanoparticle materials still faces many problems. For example, the nanoparticles often block the micropores and inhibit the free migration of lithium ions during the charging and discharging process, leading to the increase of battery internal resistance [13]. Moreover, some inorganic nanoparticles would detach from the separator surface because of the interfacial stress resulting from the manufacturing process of LIBs, which gives rise to nonuniform impedance distribution, potentially resulting in the thermal runaway of LIBs [22]. Additionally, these electrochemical inert solids (ceramics and binders) are not effective enough in the state-of-the-art research on achieving better electrochemical performance, such as higher energy density, higher power density, and so on [23,24]. Minimizing the usage amount of the electrochemical inert solids can improve the electrochemical performances, but the ultrathin layers are not effective in improving thermal stability.

Within the structure of surface-coated separators, the inorganic nanoparticle is bound onto the base film (i.e., PE membrane) via the action of a binder, which is called the “point bonding” pattern. Such a bonding pattern entails a thicker coating or higher adhesive usage to achieve remarkably reduced thermal shrinkage. Therefore, only by changing the interaction mode between the coating layer and the base film can the thermal stability of the PE membrane be substantially improved while simultaneously decreasing the use of electro-chemically inert solids. Recently, the use of one-dimensional [25] or two-dimensional nanomaterials instead of inorganic nanoparticles as the surface coating of the base film can transform the “point bonding” pattern into the “inter-line or inter-plane bonding” mode, thus greatly enhancing the adhesive force and thermal stability. For example, Zhao [26] prepared nanofibers-coated polypropylene (PP) separator with a three-dimensional network of interlacing Mg_2_B_2_O_5_ bundles, which hardly shrank after heating at 160 °C for 30 min. Hu [27] transplanted catechol functional groups on the surface of a PP separator with dopamine to interact with aryl nanofibers, and the results showed that high thermal stability could be achieved even if a small amount of aryl nanofibers were immersed (0.005% concentration emulsion). Zhang [28] decorated the PP membrane using hexagonal Mg(OH)_2_ nanosheets and found that it could maintain its original shape after heating at 180 °C for 30 min even at a coating thickness of 0.035 μm. Inspired by the aforementioned research findings, in this paper, TiO_2_ nanorods were used for the surface modification of the PE membrane. Under the condition of “wire bonding” interaction, the influence of the thickness of the modification layer on the performance of PE base film is discussed, and the balance point between the two is sought to obtain a ceramic separator of high thermal stability with little effect on battery power and energy density. 

## 2. Experimental

### 2.1. Material Preparation

The TiO_2_ nanorods were synthesized based on the previously published method [29]. The detailed preparation process is as follows: 1.0 g TiO_2_ powder (Analytical Reagent, Aladdin- Reagent) was dispersed into 200 mL of 10 M sodium hydroxide solution, well stirred, and then transferred to a Teflon-lined autoclave for hydrothermal reaction at 180 °C for 24 h. After the reaction, the resulting white precipitate was washed with dilute HCl and deionized water until the filtrate became neutral. Finally, the product was dried at 80 °C for 10 h.

### 2.2. Preparation of Ceramic Separator

The coating slurry is prepared as follows: 1.0 g PVDF is dissolved in a certain amount of NMP solvent at 50 °C to form a uniform solution, then 1.5 g TiO_2_ nanorods and 0.01 g polyvinylpyrrolidone (PVP) are added and stirring is continued to form a uniform solution. After carefully adjusting the viscosity and solid content of the suspension, the as-prepared slurry was coated on the surface of the pristine PE membrane by the doctor-blade technique to obtain a ceramic separator. After drying, the targeted ceramic separator is prepared. The ceramic separators with a designed coating amount of 0.6, 0.9, 1.2, and 1.5 mg/cm^2^ are named C-0.6, C-0.9, C-1.2, and C-1.5, respectively, and the actual coating amount is obtained by the subsequent weighing method, which is 0.6, 0.97, 1.24, and 1.83 mg/cm^2^, respectively. For comparison and analysis, the uncoated blank PE base film was named C-0. In addition, the same coating method was used to coat the PE base film with TiO_2_ nanoparticles (Aladdin reagent, about 50 nanometers in diameter). The sample is named C-P.

### 2.3. Characterization Analysis

The crystal structure of the prepared TiO_2_ material was examined with X-ray powder diffraction (XRD) using a Bruker D8 Advance diffractometer with Cu-Kα source (λ = 1.54056 Å) from 10° to 90° with a step size of 0.02° s^−1^. The surface morphology, cross-section morphology, and SEM-EDS linear scanning of the TiO_2_ material and the coated layer were investigated using a scanning electron microscope (SEM, Philip-XL30). The thermostability of the samples was measured with differential scanning calorimetry (DSC, TA-Q200) in a temperature range of 60–200 °C at a heating rate of 10 °C min^−1^ under N_2_ flow. The thermal shrinkage of the separators (original size: 3 × 3 cm) was determined by measuring their dimensional changes after storage at 140 °C for 30 min. The degree of thermal shrinkage was calculated by using the equation (Thermal shrinkage ratio (%) = (*A*_1_ − *A*_2_)/*A*_1_ × 100%), where *A*_1_ is the initial area and *A*_2_ is the final area of the separator after the storage test. The tensile properties were tested using an Instron Universal Testing machine (RGWT-4002, Reger, Shenzhen, China). At least five independent measurements were performed for each sample with a constant rate of 20 mm s^−1^, a 20 mm length. The Liquid electrolyte uptakes of the separators were measured in 1 mol L^−1^ LiPF_6_ (EC:DMC = 3:7) solution at room temperature inside the glove box for 6 h. Liquid electrolyte-soaked membranes were weighed immediately after removing the excrescent surface electrolyte by wipes. The liquid electrolyte uptakes were calculated using the equation of (*M*_1_ − *M*_0_)/*M*_0_ × 100%, where *M*_0_ and *M*_1_ were the weight of the membrane before and after immersion in the liquid electrolyte, respectively. The air permeability of separators was examined with a Gurley densimeter (UEC, 1012 A) by measuring the time for 100 cc of air to pass through under a given pressure.

The ionic conductivities of the separators were measured by electrochemical impedance spectroscopy (EIS, CHI-660 E). A total of 2025 coin-type test cells were assembled by sandwiching the separator between two stainless steel (SS) electrodes and soaking it into the liquid electrolyte (1 M LiPF_6_ in 3:7 (volume ratio) mixture of ethylene carbonate (EC) and dimethyl carbonate (DMC)) for AC impedance measurements. Impedance data were obtained in the frequency range of 1 Hz–100 kHz with an amplitude of 10 mV at room temperature. The ionic conductivity (*κ*) was calculated using the equation (*κ* = *L*/(*R* × *S*)). Here, *R* is the electrolyte resistance measured by AC impedance, and *L* and *S* are the thickness and area of the separators, respectively. The electrochemical stability window of the separators was estimated by a linear sweep voltammetry program of the CHI-660E electrochemical workstation to check oxidation decomposition, where the stainless steel was used as the working electrode and the lithium metal was used as the counter electrode at a scan rate of 10 mV s^−1^ from 2.5 V to 6.0 V versus Li/Li^+^. The interfacial resistance between lithium electrodes was determined from the AC impedance spectrum recorded for Li|separators|Li cell over storage for up to 2 days. The measurement was carried out over a frequency range of 65,000 Hz to 0.01 Hz, with an amplitude of 10 mV.

The electrochemical performance of the prepared separators was examined using 2025 coin-type cells, comprising of the prepared separators, a cathode [LiCoO_2_ (active material):polyvinylidene fluoride (PVDF, binder):Super-P (conducting agent) = 80:10:10 wt%], and a lithium foil as an anode. Then, 1 M LiPF_6_ in EC/DMC 3:7 by volume was employed as an electrolyte. All the test cells were assembled in a dry, argon-filled glove box. The assembled coin cells were charged/discharged in the voltage range of 3.0~4.3 V on the CT2001A cell testing instrument (Land Electronic Co., Ltd.) at currents of 0.2, 0.4, 1.0, 2.0, 3.0 C, 5.0 C, and 7.0 C to test the rate capability. For the cycle stability, the charge/discharge current density was fixed at 0.5 C. All electrochemical tests were conducted at room temperature. Electrochemical impedance spectroscopy (EIS) was performed on an electrochemical workstation, while the impedance spectra were recorded under a 0.02 V amplitude and a frequency range of 50 mHz~10^5^ Hz.

## 3. Results and Discussion

The crystal structure of TiO_2_ nanorods was analyzed by XRD as shown in Figure 1a. It was found that all the peaks were completely consistent with the standard diffraction peaks (JCPDS: 46–1237), indicating that the prepared TiO_2_ sample is a pure phase. And the broad diffraction peaks illustrate that its grain size is small. The SEM image of the TiO_2_ sample (Figure 1b) shows that lots of nanorods with a diameter of about 10–100 nanometers and a length of about tens of microns are uniformly dispersed. In ancient China, straw or bamboo, having similar morphology to the TiO_2_ sample, was commonly mixed with clay into a slurry and then coated on the wall surface, as shown in Figure 1c. The coating layer not only has good permeability but also binds strongly with the wall because the straw or bamboo interacts with the wall through a “wire bonding”. Therefore, if the synthesized TiO_2_ nanorods are prepared into a viscous slurry and coated on the surface of the PE separator, a protective layer with strong binding force and high permeability can also be obtained, which further improves the thermal stability of the PE separator without affecting the free migration of lithium ions.

The surface morphologies of the pristine PE separator and ceramic separators are shown in Figure 2. The pristine PE separator (Figure 2a) has a typical interconnected submicron pore structure originating from the wet method. This structure can facilitate the storage of electrolytes and allow the free migration of lithium ions inside the separator. Compared with the pristine separators, as depicted in Figure 2b,d–f, all the ceramic separators have similar surface morphologies for the inorganic coating layer, where the TiO_2_ nanorods are uniformly dispersed and interlaced on the surface of the PE separator, forming a three-dimensional network with porous structure. The surface morphology of the reverse side of the coating layer for the sample of C-0.6 was also examined, as shown in Figure 2c. It was found that the pore structures of the PE separator can be maintained after the coating process, indicating that the TiO_2_ nanorods, unlike other nanoparticles, did not clog the micropores because the nanorods with a high ratio of length to diameter tend to bridge over the micropores while the nanoparticles with a small size can be embedded into the micropores preventing the Li-ions from migrating through the separator. Besides, the morphologies of the cross-sectional and the thicknesses of the coating layer for the ceramic separators were observed with SEM and liner SEM methods, respectively, as shown in Figure 2g–j. The structures of the coating can be observed clearly, and the thickness for the samples of C-0.6, C-0.9, C-1.2, and C-1.5 are 0.5, 0.9, 1.3, and 2.1 μm, respectively. These results are consistent with the actual coating amount tested by weighing method, where the actual coating amount for sample C-0.6, C-0.9, C-1.2, and C-1.5 are 0.6, 0.97, 1.24, and 1.83 mg/cm^2^, respectively.

In order to investigate the thermal-resistant characteristics of the ceramic separators with different coating amounts, thermal shrinkage behaviors are observed by measuring the dimensional change (area-based) after storing the separators at 140 °C for 0.5 h. The results are shown in Figure 3a–g. It can be seen that the pristine PE separator (in Figure 3a) easily loses dimensional stability due to its low melting point of around 130 °C, which may cause an internal short circuit in the battery, further resulting in thermal runaway. When the synthesized TiO_2_ nanorods are used to modify the pristine PE separator, their thermal stability at high temperatures is significantly improved even under low coating amounts, as shown in Figure 3b. The thermal shrinkage of C-0.6 is about 4.5%, which is within the acceptable value (5%) for commercial cell [30,31,32]. When the coating amount increases, as shown in Figure 3c–e, their thermal shrinkage decreases gradually. The shrinkage of C-1.5 is almost negligible, while the coating amount increases to 1.83 mg/cm^2^. In addition, the morphology for the reverse side of the coating of sample C-0.6 after the thermal shrinkage test was also investigated by SEM measurement. Compared with the surface characteristics before the thermal shrinkage test (Figure 2a,c), most of the micropores can still maintain the original structure except for some micropores closed by melting in Figure 3f, indicating that the ultrathin coating layer composed of TiO_2_ nanorods can effectively inhibit the shrinkage of the pristine PE separator at high temperature. It is observed in Figure 3b–f that the optimum coating amount is about 0.6 mg/cm^2^ (C-0.6) because it will not introduce too much inert material and reduce the energy density of the battery while can obtain decent thermal shrinkage. In order to study the influence of different coating materials on the thermal stability of the separator, TiO_2_ nanoparticles with a diameter of about 50 nm were also used to modify the PE surface. The coating amount was tailored based on its thermal shrinkage, which is exactly equivalent to that of sample C-0.6, as depicted in Figure 3g,h. The coating thickness for C-P is about 4 μm which is much thicker than that of C-0.6 (0.5 μm, observed in Figure 2g). It unequivocally demonstrated that the coating material with nanorods compared with nanoparticles could not only effectively reduce the coating thickness and the use of inert substances, improve the energy density of batteries, but also effectively inhibit thermal shrinkage of the separator at high temperatures. Moreover, the differential scanning calorimeter (DSC) analysis was carried out to illustrate the effect of surface coating on the thermal stability of ceramic separators with different coating amounts. As shown in Figure 3i, the melting point of the pristine PE separator is 134.9 °C, which is consistent with previous literatures [33,34]. While the melting point of the ceramic separators was significantly improved after coating with TiO_2_ nanorods, a smaller increase was observed after the coating thickness increased to more than 1.3 μm (C-1.2 and C-1.5). Therefore, it can be concluded that the thermal stability for all the ceramic separators was greatly improved, but the optimum coating amount is about 0.6 mg/cm^2^ because when the coating amount is greater than this value, the introduction of massive non-electrochemical substances and the resulting reduction of the energy density will offset the slightly improved thermal stability of separators.

The influence of coating amount on the mechanical properties of separators is analyzed by tension testing. As shown in Figure 4a, the tensile strength of the pristine separator (sample C-0) is 15.5 MPa, and it can be improved after coating with the synthesized TiO_2_ nanorods, as evidenced by the tensile strengths of 17.0 (C-0.6), 17.36 (C-0.9), 17.98 (C-1.2) and 18.05 (C-1.5) MPa, respectively, which is consistent with the aforementioned thermal stability analysis. Therefore, based on the analysis results of mechanical properties and thermal stability, a possible mechanism was proposed in Figure 4b. The coating can be divided into a surface layer and a stacking layer. Firstly, in the surface layer, some TiO_2_ nanorods “bridged” on the skeletons of the microporous of the pristine membrane through the binder. Then, the other TiO_2_ nanorods will stack on the surface layer to form a stacking layer, where these TiO_2_ nanorods do not interact directly with the pristine membrane. When the separator is subjected to external force (mechanical stretching or thermal shrinkage), the micropores can maintain the original shape and keep themselves from thermal shrinkage or mechanical stretching mainly due to the interaction between the skeletons of the microporous and TiO_2_ nanorods in the surface layer rather than the interaction between the skeletons of the microporous and TiO_2_ nanorods in stacking layer. Therefore, smaller improvements in thermal stability and mechanical strength were observed when the coating amount of TiO_2_ nanorods increased to above 0.6 mg/cm^2^. However, if the TiO_2_ nanoparticles are used to modify the pristine separator, the interaction between TiO_2_ nanoparticles and the skeleton of the pristine separator can only occur by “point” gluing rather than “bridging” gluing when the separator is subjected to external force (mechanical stretching or thermal shrinkage), this bonding mode is very inefficient in preventing the shrinkage or extension of the separator.

The electrochemical performances of ceramic separators with different coating amounts were characterized by the electrochemical workstation. The electrochemical window is a vital parameter to evaluate the electrochemical stability of separators, which were usually investigated by linear sweep voltammetry (LSV) tests. Generally, the onset of the suddenly increasing current was caused by the oxidative reaction of electrolyte decomposition, and the corresponding voltage indicates the maximum electrochemical stable voltage [35,36]. As shown in Figure 5a, there were no obvious current changes during the potential sweeps at 4 V, whereas the current showed a dramatic difference between 4.0 and 5.5 V. The electrochemical stabilities of PE separators are about 4.1 V (vs. Li^+^/Li). In comparison, the current onsets of C-0.6, C-0.9, C-1.2, and C-1.5 separators are 5.3 V. The TiO_2_ nanorods-coated separator has a wider electrochemical stability window, which indicates that the TiO_2_ nanorod-modified separator possesses better electrochemical stability. The stability enhancement means better compatibility with the electrolyte of the lithium-ion battery, which should be attributed to the excellent electrolyte affinity of TiO_2_-coated PE separator and the stabilization of electrolyte anions by Ti-O units acting as the Lewis acid centers [37,38,39]. Ionic conductivity is another important indicator to evaluate the electrochemical performance of the separator. Figure 5b shows the Nyquist plots of the stainless steel (SS)/separator-electrolyte/SS molds assembled by sandwiching the pristine PE separator or coated separators soaking in liquid electrolyte between two pieces of SS. The high-frequency intercept on the real axis reflects the bulk resistance (*R*_b_), which can be used to calculate the ionic conductivity in Table 1. According to the results, all TiO_2_ nanorod-modified PE separators show higher ionic conductivity than the pristine PE separator, and the highest ionic conductivity was observed for sample C-0.6, which may benefit from the synergistic contributions of the significantly increased electrolyte uptake and the well-preserved porous structure [13,40], as observed in Figure 2c. The compatibility of liquid electrolyte-soaked separators with a lithium electrode is also a very important factor in the C-rate capability of lithium-ion batteries, which can be investigated by evaluating the impedance variation of Li/liquid electrolyte-soaked separator/Li cells. As shown in Figure 5c, a semicircle was observed from the impedance spectra of cells with all separators that represent the Li/electrolyte interfacial resistance (*R*_int_), which was related to the charge transport across the passivation layer (solid electrolyte film) and the charge transfer reaction, Li^+^ + e^−^ = Li [3,37]. The *R*_int_ for C-0, C-0.6, C-0.9, C-1.2, and C-1.5 samples are 243 Ω, 118 Ω, 124 Ω, 141 Ω and 161 Ω, respectively. Compared with the uncoated PE separator, it can be seen that all TiO_2_ nanorod-modified PE separators had lower interfacial resistance, indicating smooth ion transport between the ceramic nanoparticle-coated separators and electrodes. It can be attributed to the TiO_2_ nanorod-modified PE separators capable of retaining the original porous structure and the layer of TiO_2_ nanorods able to obtain higher electrolyte uptake (Table 1), which can effectively decrease the interaction between electrolyte components and the lithium electrode, and gradually stabilizes the interface [41]. Moreover, the C-0.6 separator exhibited the smallest interfacial impedance and therefore had the best separator-electrode compatibility. This is consistent with previous research [38,42] that a very thin inorganic oxide layer can negate the interfacial impedance between the electrolyte and lithium metal, owing to the high binding energy between lithium and the oxides layer.

From the results of thermal stability, mechanical properties, and electrochemical performance tests for TiO_2_ nanorod-modified separators with different coating amounts, sample C-0.6 has the best overall performance, such as the best ionic conductivity and interfacial impedance, excellent electrochemical stability window, acceptable thermal stability and mechanical properties, the minimum introduction of inert ingredients. Therefore, the electrochemical performance of the half-cell composed of the sample C-0.6 separator, a negative electrode (lithium metal), and a positive electrode (LiCoO_2_) was further studied. For better comparative analysis, the electrochemical performance of the half-cell assembled by pristine PE membrane (C-0) was also tested. As shown in Figure 6a, the discharge specific capacity at low rates and its capacity recovering property (0.2 C) after 7 C for C-0.6 and C-0 are very close, but their differences gradually become larger with the increase of discharge current density. When the discharge current density is 7 C, the specific capacities of C-0.6 and C-0 are ~86.4 and ~60.4 mAh g^−1^, respectively. Moreover, from their discharge curve in Figure 6b,c, it can be further seen that there is little difference in the curve shape and the average discharge voltage platform under low rates, while the voltage platform decrease for the C-0 sample is more than that of C-0.6 at high rates. These results indicate that the polarization resistance of the battery assembled by TiO_2_ nanorod-modified separator is smaller than that of the battery composed of the pristine separator at high rates, which is consistent with the test results of ionic conductivity (Figure 5b), interfacial impedance (Figure 5c) and electrolyte uptake (Table 1). Furthermore, the cycle performance of these half-cells was studied, as depicted in Figure 6d. At the discharge current density of 0.5 C, the capacity retention of the C-0.6 sample is 82.6% after 100 cycles, while that of the C-0 sample is only 77.8%. Similarly, from their representative discharge curves (Figure 6e,f), it can be seen that the C-0.6 sample can maintain a good discharge capacity after cycling.

In order to further understand the influence of the TiO_2_ nanorod-modified layer on the electrochemical performance of the PE separator, AC impedance spectrum analysis was performed on the half-cell assembled by the C-0.6 and C-0 separators. As can be seen from the Nyquist plot in Figure 7, the semicircle in the high-middle frequency region represents the interfacial resistance (*R*_int_) modified by the combined effect of solid-electrolyte interface resistance (*R*_SEI_) and charge-transfer resistance (*R*_ct_). The TiO_2_ nanorod-modified PE separator (C-0.6) exhibited a smaller semicircle at high-middle frequency, indicating that their interfacial resistance is relatively low compared with that of the pristine PE separator. These results were ascribed to surface modification leading to enhanced hydrophilicity and affinity with electrolyte, resulting in thinner and more compact SEI formation by TiO_2_ nanorod-modified PE separator, which agreed with the findings of the ionic conductivity (Figure 5b), interfacial resistance (Figure 5c) and electrolyte uptake (Table 1).

## 4. Conclusions

In this paper, the TiO_2_ nanorods with a diameter of about 10–100 nanometers and a length of about tens of microns are used to modify the PE separator. It can be observed from SEM tests that the TiO_2_ nanorods are “bridged” on the microporous skeleton of pristine PE separator to form a coating structure with a three-dimensional porous network, while the TiO_2_ nanoparticles are “embedded” into the micropores, which will cause a series of problems such as micropore blockage, easy detaching, and introduction of excessive inert substances. Then, multiple analytical techniques (e.g., SEM, DSC, EIS, LSV, and so on.) are also utilized to investigate the effect of coating amount on the physicochemical and electrochemical properties of ceramic separator. The results showed that these properties can be effectively improved by coating TiO_2_ nanorods, but the degree of improvement is not directly proportional to the coating amount. For example, the thermal stability and mechanical properties of the ceramic separator increase with the increase of the coating amount, however, smaller improvements are observed when the coating amount is above 0.6 mg/cm^2^. In fact, when the separator is subjected to external force (mechanical stretching or thermal contraction), the forces inhibiting micropore deformation are derived from the interaction of TiO_2_ nanorods directly “bridging” with the microporous skeleton rather than those indirectly “glued” with the microporous skeleton. In addition, when the loading level is 0.6 mg/cm^2^, the ceramic separator can achieve optimal performance in terms of ionic conductivity, electrochemical stability window, and interface compatibility because the introduction of excessive inert coating material can reduce the ionic conductivity, increase the interfacial impedance, and lower the energy density of the battery. Moreover, the capacity retention assembled by the ceramic separator with a loading of 0.6 mg/cm^2^ TiO_2_ nanorods was 57.1% under 7 C/0.2 C and 82.6% after 100 cycles, respectively, indicating that the ceramic separator with a thin coating layer has well-balanced performances. This research may provide a novel approach to overcoming the common disadvantages of current surface-coated separators.

## Figures and Tables

**Figure 1 materials-16-02049-f001:**
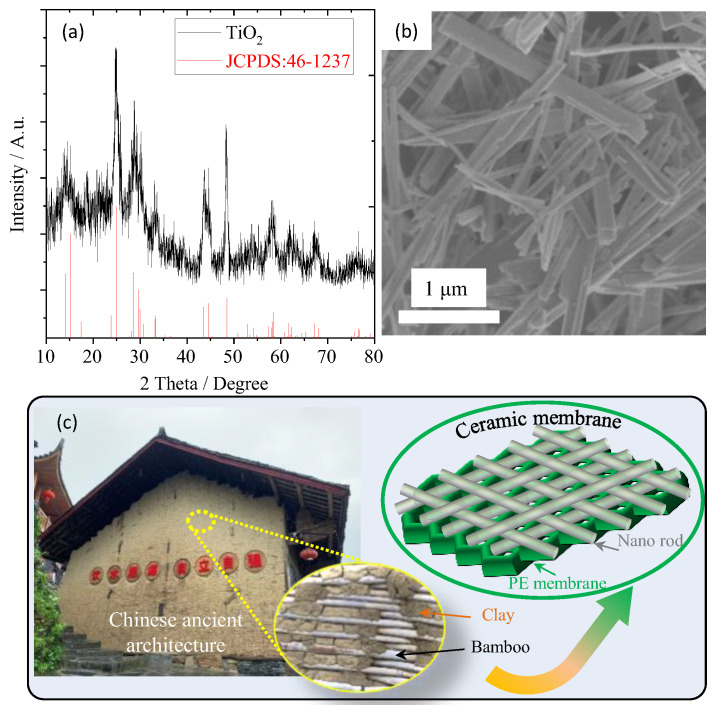
(**a**) XRD pattern and (**b**) SEM image of the as-synthesized TiO_2_ nanorods, (**c**) is the schematic diagram for ceramic separator obtained from TiO_2_ nanorods.

**Figure 2 materials-16-02049-f002:**
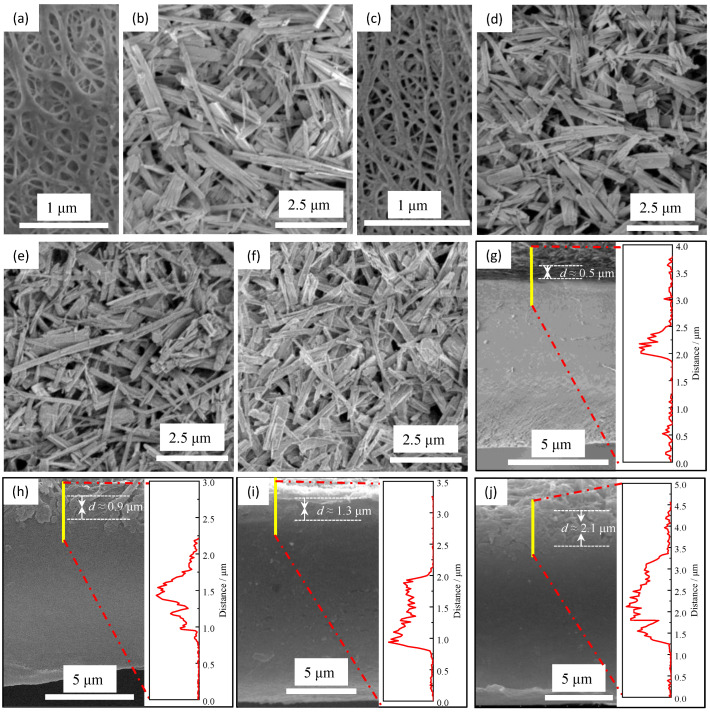
The SEM images for the pristine PE separator (**a**) and the ceramic separators with different coating amounts (**b**–**j**): the surface morphologies for the coating layer (**b**) and the reverse side of the coating layer (**c**) of sample C-0.6. (**d**–**f**) are the surface morphologies for coating of samples C-0.9, C-1.2, and C-1.5, respectively. (**g**–**j**) are the cross-section morphologies and the thicknesses of the coating layer for the ceramic separators C-0.6, C-0.9, C-1.2, and C-1.5, respectively.

**Figure 3 materials-16-02049-f003:**
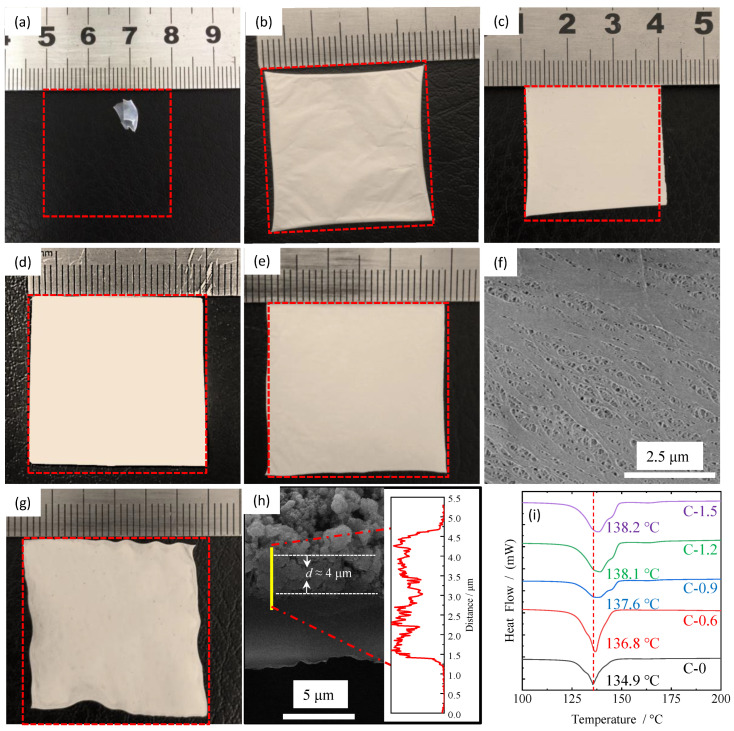
Thermal shrinkage (%) of PE separator (**a**) and ceramic separators at 140 °C for 0.5 h: (**b**) C-0.6, (**c**) C-0.9, (**d**) C-1.2, (**e**) C-1.5 and (**g**) C-P. (**f**) is the SEM image for the reverse side of the coating of sample C-0.6 after thermal shrinkage test. (**h**) is the cross-section morphology and the thicknesses of the coating layer for C-0.6. (**i**) is the DSC curves for PE separators and ceramic separators.

**Figure 4 materials-16-02049-f004:**
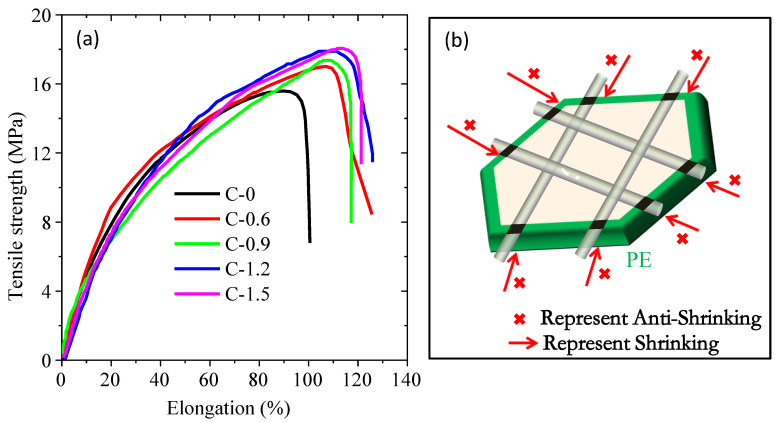
The stress-strain curves of the ceramic separators with different coating amounts (**a**) and anti-shrinkage mechanism diagram (**b**).

**Figure 5 materials-16-02049-f005:**
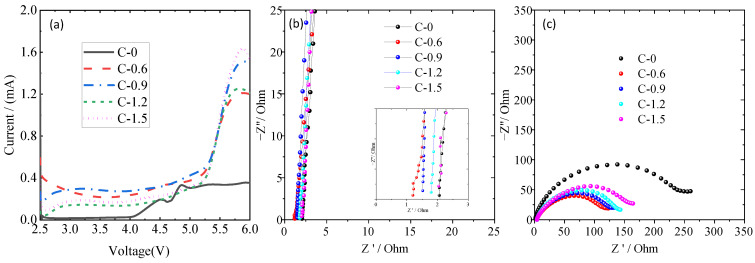
The Linear scan voltammetry (LSV) curve of Li/separator- liquid electrolyte/SS cells (**a**), AC impedance spectra of the SS/separator-liquid electrolyte/SS cell (**b**), the interfacial resistances of the Li/separator-liquid electrolyte/Li cell (**c**).

**Figure 6 materials-16-02049-f006:**
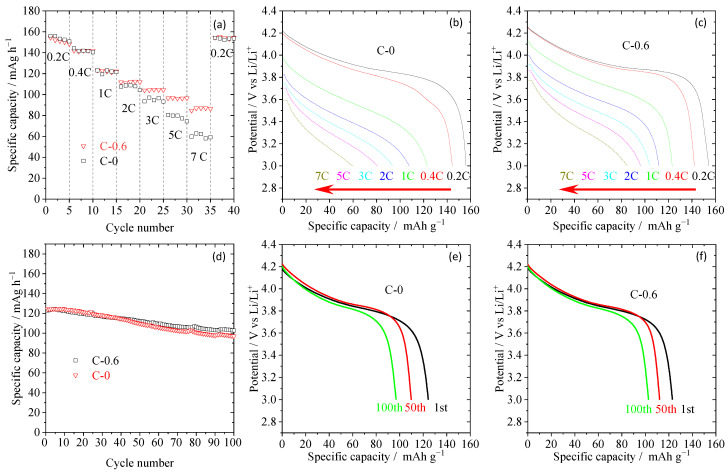
The C-rate capabilities (**a**) and their corresponding discharge profiles (**b**,**c**) of half-cells assembled with pristine PE separator (C-0 sample) and TiO_2_ nanorod-modified PE separator (C-0.6 sample), where charge/discharge current densities are varied from 0.2/0.2–7/7 C under a voltage range between 3.0 and 4.2 V. the cyclic performance at 0.5 C (**d**) and their corresponding discharge profiles (**e**,**f**) of half-cells assembled with pristine PE separator (C-0 sample) and TiO_2_ nanorod-modified PE separator.

**Figure 7 materials-16-02049-f007:**
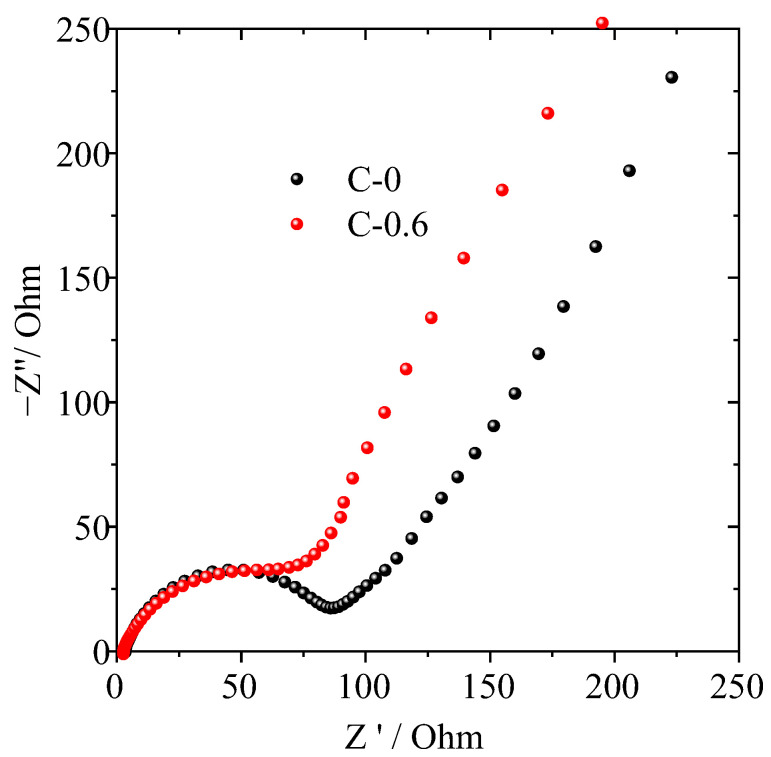
Nyquist plots of half-cells assembled with pristine PE (C-0) and TiO_2_ nanorod-modified PE (C-0.6) separator.

**Table 1 materials-16-02049-t001:** Properties of the surface-modified separators.

Sample	Thickness of Coating (μm)	Gurley Value (s 100 cc^−1^)	Electrolyte Uptake (%)	Thermal Dimensional Shrinkage (%,140 °C)	Melting Temperature (°C)	Ionic Conductivity (mS cm^−1^)
C-0	0	178	132	~98.0	134.9	0.33
C-0.6	~0.5	191	269	~10.0	136.8	0.57
C-0.9	~0.9	213	283	~5.65	137.6	0.47
C-1.2	~1.3	226	299	~5.69	138.1	0.40
C-1.5	~2.1	241	311	~3.0	138.2	0.38

## Data Availability

Not applicable.

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
