# Peer review of "TiO2 Nanorod-Coated Polyethylene Separator with Well-Balanced Performance for Lithium-Ion Batteries"

_materials, 2023, doi:10.3390/ma16052049_

Round 1

Reviewer 1 Report

The authors studied the effect of TiO2 coated separator on the performance of mechanical, thermal and electrochemical performance. The study can help to design the separator for better performance of batteries. The following discussion can be helpful for clear understanding.

1. C-0.6 shows the best ionic conductivity in table 1. The reason is addressed as the increased electrolyte uptake and well-preserved porous structure. What the mechanisms of better uptake for TiO2 coated surface is?

2. As TiO2 increases, electrolyte uptake increases but porosity decreases. Thus, ionic conductivity is a trade-off of uptake and porosity. For the case of smaller amount of 0.6, the uptake decreases, and porosity increases so the ionic conductivity can increase but there can be a point the maximum ionic conductivity is achieved. More discussion on the range of below 0.6 is needed. What will be the maximum ionic conductivity by the TiO2 coating method? Is there any bottom amount of TiO2 for better performance?

3. The effect of TiO2 particle is shown only for thermal performance. What the effect of TiO2 particle on electrochemical performance is? It will be interesting to compare ionic conductivity for various amount of TiO2 particle coating with TiO2 rod coating. 

Author Response

Question 1. C-0.6 shows the best ionic conductivity in table 1. The reason is addressed as the increased electrolyte uptake and well-preserved porous structure. What the mechanisms of better uptake for TiO2 coated surface is?

Answer: The possible mechanism is due to its unique coating structure. Compared with the pristine separators, as depicted in Fig. 2b and Fig. 2(d-f), the TiO2 nanorods for the ceramic separators are uniformly dispersed and interlaced on the surface of PE separator forming a three-dimensional network with porous structure. This structure can facilitate the storage of electrolyte and allow the free migration of lithium ions inside the separator.

Question 2. As TiO2 increases, electrolyte uptake increases but porosity decreases. Thus, ionic conductivity is a trade-off of uptake and porosity. For the case of smaller amount of 0.6, the uptake decreases, and porosity increases so the ionic conductivity can increase but there can be a point the maximum ionic conductivity is achieved. More discussion on the range of below 0.6 is needed. What will be the maximum ionic conductivity by the TiO2 coating method? Is there any bottom amount of TiO2 for better performance?

Answer: Ionic conductivity is a trade-off of uptake and porosity. Therefore, your suggestion is very good. However, in order to obtain the case of smaller amount of 0.6, the coating method needs be changed, such as sputtering. In this paper, doctor-blade technique was used to obtain ceramic separator, which is also commonly used in industries. However, the disadvantage of this method is that it is difficult to control the coating with thin thickness, so the range below 0.6 was not discussed in this study.

Question 3. The effect of TiO2 particle is shown only for thermal performance. What the effect of TiO2 particle on electrochemical performance is? It will be interesting to compare ionic conductivity for various amount of TiO2 particle coating with TiO2 rod coating.

Answer: Thank you for your suggestions. It is interesting to compare ionic conductivity for various amount of TiO2 particle coating with TiO2 rod coating. However, the coating thickness for C-P (TiO2 particle) is much thicker than that of C-0.6. From the point of view of practical application, these electrochemical inert solids (ceramics and binders) are not conducive to obtaining higher energy density and higher power density. Therefore, minimizing the usage amount of the electrochemical inert solids is essential. In this paper, our purpose is to discuss the influence of the thickness of the TiO2 rod modification layer on the performance of PE base film with doctor-blade technique, and the balance point between the two is sought to obtain ceramic separator of high thermal stability with little effect on battery power and energy density. So, the effect of TiO2 particle on electrochemical performance is not compared in this paper. We will take your suggestions into consideration in the following research.

Reviewer 2 Report

In this work, TiO2 nanorods were used to enhance commercial PE separators that can improve the safety and electrochemical performance of lithium batteries. The safety issue of lithium batteries has been attracting increasing attention while the energy capacity becomes larger in order to meet the practical application (power density and long-term cycling). So, this work shows a promising strategy to improve the safety of current widely used polyolefin separators, which is quite interesting. Below are some comments made to hopefully perfect this work.

- To make the TiO2 clearer, TEM images should be provided.

- In Figure 5a, authors should check the data of LSV, as in my experience, the curve of C-0 should be smooth and the value should not be as large as the one shown in Figure 5a. if too large, it means there are extra redox reactions inside of batteries.

- In Fig 6a, authors should supplement the data from 7C recovering to 0.2 C to evaluate the performance when there is a huge current density change. 

- As this work introduces a one-dimensional nanoparticle TiO2 to enhance the safety of lithium batteries, here some papers are recommended to cite and probably can improve this work as well. Chemical Engineering Journal 451, 138496; Applied Materials Today 21, 100793; ACS Applied Energy Materials 2 (6), 4167-4174; Journal of Materials Chemistry A 7 (12), 6859-6868.

Author Response

Question 1. To make the TiO2 clearer, TEM images should be provided.

Answer:  The TiO2 nanorods were synthesized based on the previously published method (Journal of Crystal Growth, 2010, 312, 213-219), which was a work of our previous group (Prof. Wei is my supervisor). In addition, the morphology of the TiO2 nanorods was verified by SEM test in Fig. 1b. Therefore, the TEM test was not conducted in this paper. Moreover, we were given only 5 days to update the revision, and it was difficult to complete the TEM test in a short time. So, we did not supplement the TEM data in the revision.

Question 2. In Figure 5a, authors should check the data of LSV, as in my experience, the curve of C-0 should be smooth and the value should not be as large as the one shown in Figure 5a. if too large, it means there are extra redox reactions inside of batteries.

Answer: We have conducted the LSV test again to confirm the accuracy of C-0. It can be seen from the  figure that a similar curve was obtained. The peaks indeed are larger than previous studies, which could be attributed to the side reaction between some functional groups of separator and electrolyte. This phenomenon has not been observed in our ceramic separators, illustrating that our improvement for PE separator by TiO2 nanorods is effective.

Question 3. In Fig 6a, authors should supplement the data from 7C recovering to 0.2 C to evaluate the performance when there is a huge current density change. 

Answer:  The data was supplement in our revisions as can be seen in Fig.6a.

Question 4. As this work introduces a one-dimensional nanoparticle TiO2 to enhance the safety of lithium batteries, here some papers are recommended to cite and probably can improve this work as well. Chemical Engineering Journal 451, 138496; Applied Materials Today 21, 100793; ACS Applied Energy Materials 2 (6), 4167-4174; Journal of Materials Chemistry A 7 (12), 6859-6868.

Answer:  Thank you for your suggestions. These references were supplement in our revisions, such as number 6, 12 and 25.
